# A common computational principle for vibrotactile pitch perception in mouse and human

Mario Prsa [1✉], Deniz Kilicel[2], Ali Nourizonoz[2], Kuo-Sheng Lee[2] & Daniel Huber [2✉]

We live surrounded by vibrations generated by moving objects. These oscillatory stimuli propagate through solid substrates, are sensed by mechanoreceptors in our body and give rise to perceptual attributes such as vibrotactile pitch (i.e. the perception of how high or low a vibration's frequency is). Here, we establish a mechanistic relationship between vibrotactile pitch perception and the physical properties of vibrations using behavioral tasks, in which vibratory stimuli were delivered to the human fingertip or the mouse forelimb. The resulting perceptual reports were analyzed with a model demonstrating that physically different combinations of vibration frequencies and amplitudes can produce equal pitch perception. We found that the perceptually indistinguishable but physically different stimuli follow a common computational principle in mouse and human. It dictates that vibrotactile pitch perception is shifted with increases in amplitude toward the frequency of highest vibrotactile sensitivity. These findings suggest the existence of a fundamental relationship between the seemingly unrelated concepts of spectral sensitivity and pitch perception.

---

[1] Department of Neuroscience and Movement Science, University of Fribourg, Fribourg, Switzerland. [2] Department of Basic Neurosciences, University of Geneva, Geneva, Switzerland. ✉email: mario.prsa@unifr.ch; daniel.huber@unige.ch

Pallesthesia is the clinical term to designate the sense of vibrations. In clinical practice, physicians test pallesthesia in their patients by applying a vibrating tuning fork against bones of lower and upper limbs. Indeed, Pacinian corpuscles, the mechanoreceptors specialized in transducing high frequency (>100 Hz) vibrations, can be found deep inside the forearm adjacent to joints and bones[1,2]. In turn, their innervating primary afferent neurons, located in the dorsal root ganglia, transmit the information along the ascending neuraxis to the somatosensory cortex, allowing us to consciously perceive properties of the vibratory stimulus. In the auditory system, airborne vibrations (i.e., sound) can evoke pitch perception, making it possible to distinguish for example high from low notes or voices. It is quantified on a frequency scale but is a function of several physical properties of sound[3]. Similarly, vibrotactile pitch perception is perhaps what allows one to identify the source of a nearby movement, such as a large or small object, a conspecific, a predator or a prey[4–7]. Despite its importance, a systemic quantitative assessment of this percept is currently lacking in the somatosensory literature.

On the one hand, standard V-shaped sensitivity curves have been established in humans and non-human primates[8,9], and show that maximal vibration sensitivity occurs around 240 Hz. On the other, some evidence exists that human subjects can perceive that a vibration has higher or lower pitch than a frequency-matched comparison if its amplitude is changed[10]. Our previous work indeed suggests that vibrotactile pitch perception in mice seems to be a complex function of multiple physical stimulus attributes, such as frequency and amplitude[2]. Can this function be precisely quantified, is it universal across species and is it in any way related to the spectral sensitivity curve? To answer these questions, we trained mice and humans in a frequency discrimination task at multiple spectral locations and tested if and how changes in vibration amplitude affect their perceptual responses.

Our findings reveal that vibrotactile pitch is computed as a product of stimulus frequency and a power function of its amplitude in both mouse and human. The value of the power function exponent depends on the spectral location relative to the frequency of highest vibrotactile sensitivity.

## Results

We trained mice, using a go/no-go task design (Fig. 1A), to discriminate the frequency of vibrotactile stimuli (pure sinusoids) applied to their forelimb. Four high frequencies (go response) had to be distinguished from four low frequencies (no-go response) uniformly distributed around a center frequency. This experiment was repeated at three different spectral locations (center frequencies at 450, 1000, and 1600 Hz). Mice were able to learn the discrimination task and perceived the stimuli on a continuum, as evidenced by the psychometric curve fits to their perceptual responses (Fig. 1B, black traces). We then reasoned that if pitch perception depends exclusively on vibration frequency, their responses should not be affected when vibration amplitude is changed. To test the effect of amplitude change, after being trained on the frequency discrimination task at a fixed reference amplitude for 12 consecutive days, we introduced different probe amplitudes on 30% of the trials, each on a separate day. At the 450 Hz spectral location, the amplitude change consistently shifted the psychometric curves: an amplitude increase required a decrease in stimulus frequency, and vice versa, in order to evoke the same perceptual response (Fig. 1B, colored traces). By fitting the frequency shift ratio as a function of the amplitude change factor (ACF), we identified that vibrotactile pitch is expressed as the product of vibration frequency ($f$) and a power function of

vibration amplitude ($A$), two independent physical attributes (Fig. 1C). The $A^k \times f$ curve, with k = 0.32 (fit to the data of four mice), represents all amplitude/frequency pairs that evoke the same pitch percept as a 450 Hz vibration at the 5.6 μm reference amplitude (black square in Fig. 1C). Probe amplitudes did not affect frequency discrimination at the 1000 Hz spectral location (k not significantly different from 0) and yielded a negative k exponent (k = −0.044) for vibrations at the 1600 Hz location (Fig. 1C).

We next asked whether the same rule governs vibrotactile pitch perception in humans. Participants were instructed to compare the perceived frequency (and ignore the amplitude) of two consecutive vibrations (a test and a standard) delivered to the fingertip of their index finger in a two-alternative forced choice task design (Fig. 2A, see Methods for details). The standard stimulus was either a 160, 200, 280, 440, or a 480 Hz vibration, and the frequency of the test stimuli uniformly distributed around the standard. The four standard stimuli were each tested on a different day. Test stimuli were always presented at the same amplitude of 11.8 μm and the standard was presented at seven different reference amplitudes. As in the mouse data, changing vibration amplitude consistently shifted the psychometric curves so that pitch can be expressed as the $A^k \times f$ product (Fig. 2B,C). However, the fitted k exponent was negative (k = −0.24, fit to the median of nine subjects) for the 440 Hz standard whereas it was positive for mice at the same spectral location; meaning that a relative amplitude increase (of the test relative to the standard, ACF > 1) required an increase of vibration frequency in order to evoke the same percept. We found that the equal-pitch curves sloped negatively (k > 0) for 160 Hz and 200 Hz vibrations, and positively (k < 0) for the 280 and 480 Hz vibrations (Fig. 2C). Therefore, in both species, the k exponent changes from positive to negative as we move higher in the vibration spectrum. The transition seems to occur at 1000 Hz in mice and ≈240 Hz in humans.

To understand the significance of these transition points, we sought to establish the V-shaped sensitivity curves in mice and humans. Both were trained for this purpose in a two-alternative forced choice task. Mice had to identify the presence or absence of a vibrotactile stimulation by licking either toward a left or right reward spout, and humans had to report in which of two successive intervals a vibratory stimulus was present (see Methods for details). The detection tasks yielded comprehensive sensitivity curves (Fig. 3), which revealed that the 1000 Hz and ≈240 Hz transition points are also the frequencies of highest vibrotactile sensitivity in the mouse and human, respectively. Therefore, the difference in pitch perception of a 440/450 Hz vibration between mice and humans is relatable to this frequency being in the lower end of the perceptual range of mice and in the higher end of that of humans.

Because the perceived intensity of a vibration also depends on both amplitude and frequency, it is important to disentangle equal-pitch from equal-intensity perception. To this end, we conducted the converse experiment, using the same task design, in which participants were instructed to compare the amplitude (and ignore the frequency) of a standard and a test vibration. The standard stimulus was this time always at a fixed amplitude (6, 8, 10, or 12 μm tested in different sessions) and the amplitude of the test stimuli uniformly distributed around this standard value. Within each session, we probed seven different reference frequencies for the standard vibration whereas the test stimuli were presented at the same 200 Hz frequency. As previously, by quantifying the shift in the psychometric fits (along the amplitude axis) caused by frequency changes of the standard yielded equal-intensity curves (Fig. 4A). The amplitude/frequency pairs falling on each curve are perceived to be equally intense as the reference

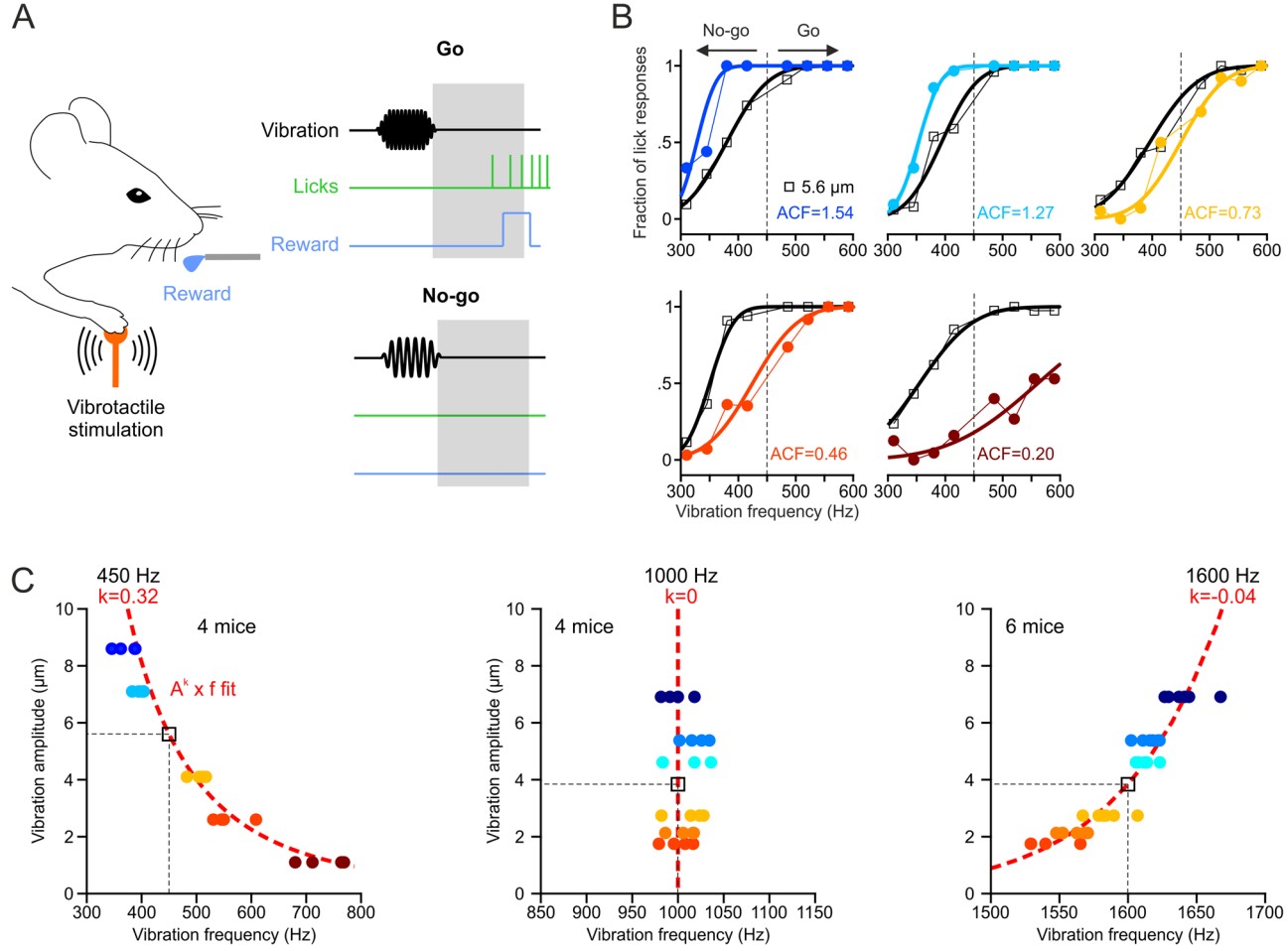

**Fig. 1 Vibrotactile pitch perception in mice. A** Schematic of the Go/No-go frequency discrimination task in mice (see Methods for details). **B** Psychometric curve fits to the fraction of Go responses for the reference 5.6 µm (black, $A_{REF}$) and probe amplitudes (colors, $A_{PROBE}$) of five test sessions for an example mouse tested at the 450 Hz center frequency. The amplitude change factor (ACF = $A_{PROBE}/A_{REF}$) is indicated for each session. **C** $A^k \times f$ equal-pitch curve fits (red lines) to vibration amplitude as a function of the frequency shift ratio (normalized to the center frequency, black square, see Methods for details) for 450 Hz ($N = 4$ mice), 1000 Hz ($N = 4$ mice), and 1600 Hz ($N = 6$ mice) center frequencies (colored symbols). Source data are provided as a Source Data file.

200 Hz vibration at the corresponding standard amplitude (black squares in Fig. 4A). The minima at ≈250 Hz confirm this to be the frequency of maximal vibrotactile sensitivity in humans and an overlay with equal-pitch curves (Fig. 4B) indicate that vibrotactile pitch and intensity are two ostensibly different perceptual phenomena.

As a final control test, to ensure that the observed perceptual shifts in the frequency discrimination task (Figs. 1B, 2B) were not guided by differences in perceived intensity of the tested vibrations, we trained participants to discriminate stimuli that all lay on the same equal-intensity curves. The standard stimulus was a 440 Hz vibration and the frequency of the test stimuli uniformly distributed between 352 and 528 Hz. The amplitude of each vibration was obtained from the 10 µm green curve in Fig. 4A (Supplementary Fig. 1A) so that all stimuli are matched in perceived intensity (i.e., are perceived equally intense as a 200 Hz vibration at 10 µm). As previously, when the amplitude of the standard was changed (ACF = 0.77 and 1.67) on a subset of trials, the psychometric curves showed consistent shifts in the expected direction along the frequency axis (Supplementary Fig. 1B). The same result was obtained when we repeated the experiment along the 12 µm equal-intensity curve (Supplementary Fig. 1C). We conclude that changes in vibration amplitude influence pitch perception independently of perceived intensity.

## Discussion

Pacinian corpuscles densely populate dermal and hypodermal layers of glabrous skin in fingertips and palm of the primate hand[11], but can only be found in deep tissue adjacent to bones in mice; mainly in their forearm, rarely in fingers and never in the palm[2]. We conclude that vibrotactile pitch perception follows a common computational principle across different mammalian species in spite of fundamentally different anatomical distribution of the involved mechanoreceptors between primate and mouse hands. This perceptual quantity is expressed in terms of a vibration's physical attributes, frequency and amplitude, as $A^k \times f$. The latter product represents perceptual constancy or metamers, that is, equal-pitch curves composed of physically different stimuli. The k exponent is adjusted so that the equal-pitch curves always slope towards the frequency of maximal sensitivity (Fig. 2C). In other words, if the amplitude of a vibration is changed by a factor N, its frequency must be shifted by a factor of $(1/N)^k$ in order to maintain the same pitch percept. The k exponent for a given equal-pitch curve is such that decreases in amplitude always require a shift along the frequency axis toward the center of the perceptual range. If the frequency is however kept constant, perception will move to a new iso-pitch curve that is closer to the range center in the case of an amplitude increase, and further from the center in the case of an amplitude decrease.

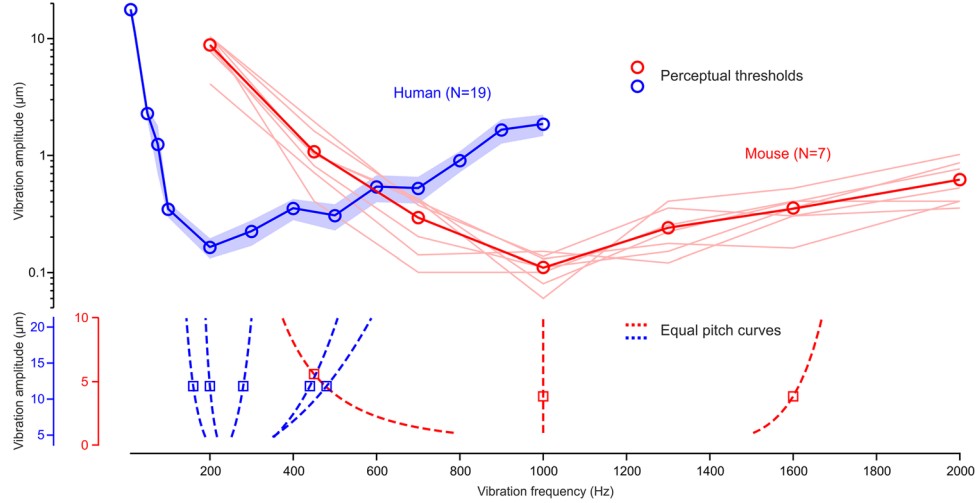

**Fig. 2 Vibrotactile pitch perception in humans. A** Schematic of the two-alternative forced choice (2AFC) frequency discrimination task in humans. **B** Psychometric curve fit to the fraction of "higher" responses for vibrations with equal reference ($A_{REF}$) and test amplitudes ($A_{TEST}$) at 11.8 μm (black), and for 6 tested amplitude change factors (ACF = $A_{PROBE}/A_{REF}$) of an example subject tested at the 440 Hz reference frequency. **C** $A^k \times f$ equal-pitch curve fits (blue lines) to vibration amplitude as a function of the median (error bars: ±quartiles, colored symbols) frequency shift ratio (normalized to the reference frequency, black square, see Methods for details) of $n = 9$ subjects, for 160, 200, 280, 440, and 480 Hz reference frequencies. Source data are provided as a Source Data file.

**Fig. 3 V-shaped perceptual sensitivity curves.** Amplitude thresholds as a function of vibration frequency for mouse (shaded lines: individual mice, symbols: mean) and human (mean ± s.e.m.). The equal-pitch curves for all tested center/reference frequencies are replotted in the bottom panel illustrating that vibratory pitch perception shifts, with increases in amplitude, toward the frequency of highest vibrotactile sensitivity in both mouse (red) and human (blue).

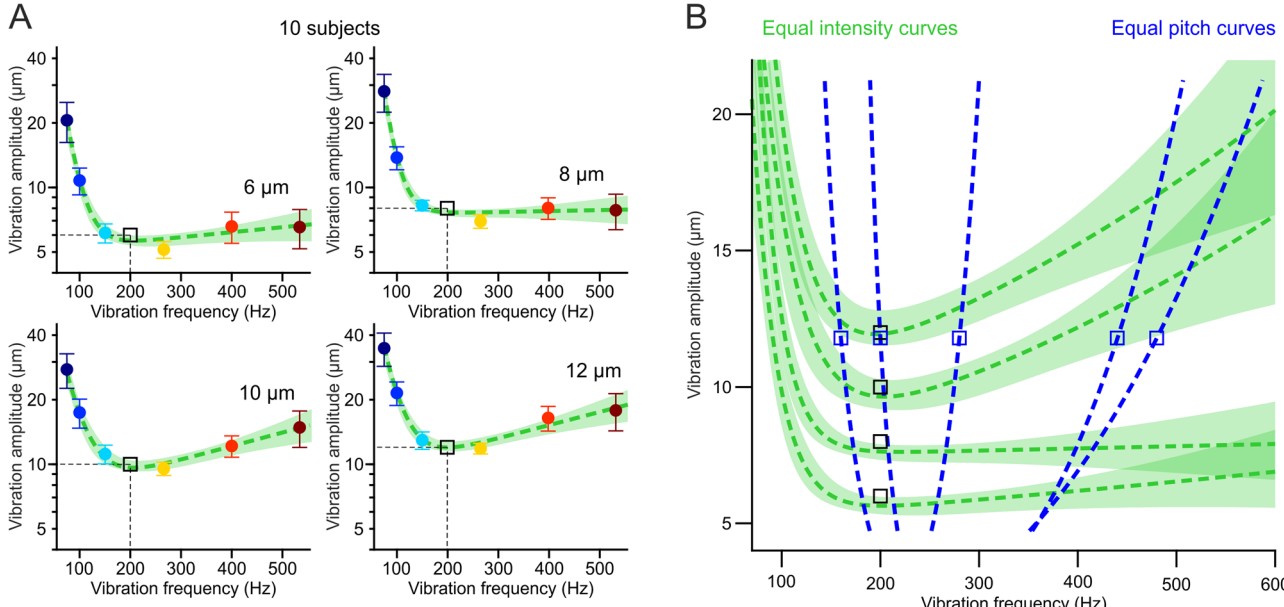

**Fig. 4 Equal-intensity and equal-pitch curves quantify two different perceptual phenomena. A** equal-intensity curves as descriptive sum of exponential fits (green lines, ±bootstrap SD) to the mean (±SEM, colored symbols) amplitude shift ratios (normalized to the reference amplitude, black square, see Methods for details) as a function of vibration frequency of $n = 10$ subjects, for 6, 8, 10, or 12 μm reference amplitudes tested in different sessions (the four panels). **B** Overlay of equal-intensity curves (from **A**) and equal-pitch curves (from Fig. 2B) show that perceptual constancy relative to a reference vibration (square symbols) follows a different rule for intensity and pitch. Source data are provided as a Source Data file.

Previous behavioral studies also reported that both humans[10,12] and rodents[13] might be blind to the physical attributes A and f of a vibration but instead perceive a composite feature. The feature was identified as the product $A \times f$ when rats were trained to discriminate a 37.5 Hz from a 75 Hz whisker vibration at two different amplitudes[13]. This is consistent with our model of a $A^k \times f$ iso-pitch curve given that the value of the $k$ exponent increases as we move lower in the vibration spectrum (Fig. 2A,B) and might thus approach unity below 100 Hz. This study however concluded that vibrations are also sensed as the $A \times f$ product when the rats were first trained to discriminate between the two different amplitudes instead of frequencies. It might in fact be impossible to disentangle pitch from intensity perception when testing very low frequencies (Fig. 3B), but the distinction becomes clear closer to the center of the vibrotactile spectrum. In contrast to these earlier reports, our psychometric approach not only allowed us to obtain a precise quantification of vibrotactile pitch perception across the whole physiological spectral range, but also reveal its underlying computational principle by linking it to the spectral sensitivity of the somatosensory system.

A similar principle seems to apply to auditory stimuli (i.e., airborne vibrations) as well. Indeed, Stevens described that changes in sound amplitude affect how high or low the pitch of a tone is perceived[14,15]. His classical work on this psychoacoustic effect also shows that iso-pitch curves slope toward the center of the hearing range, although the size of the effect seems to be much smaller than initially reported[14,15]. Recently, neural recordings revealed that in the somatosensory cortex, frequency-tuned neuronal response curves shift with changes in stimulus amplitude according to the same computational principle[2]. Similarly, when using pure tones the best frequency of an auditory cortical neuron shifts with sound attenuation toward the center of the perceptual range[16]. This effect seems to originate from the sensory periphery. On the one hand, the location of cochlear maximum excitation has been reported to shift with sound level[17], and on the other, in rapidly adapting afferents

innervating the hand, the vibration frequency that entrains the maximal number of spikes is observed to become higher for smaller amplitudes[18]. The idea that a similar computational principle governs the neural representation (and possibly perception) of pure auditory tones and sinusoidal substrate vibrations is intriguing given that the two emerge from fundamentally different sensory receptors (hair cells vs. lamellar corpuscles). Actually, it has been proposed that communication via airborne sounds might have evolved from the more ancient precursor modality based on substrate-borne vibration signaling[5]. Many insect species communicate exclusively by emitting and sensing substrate vibrations[19] while in others, the same sensory organ, such as the Johnston's organ in drosophila, is used to detect both sound and touch[20]. Vestiges of this modality seem to be still present in rodents, given that Ehrenberg's mole-rats vibrate their subterranean tunnels to communicate with conspecifics[21,22], and might explain the parallels between frequency representation in auditory and somatosensory systems.

## Methods

**Mice**. All experiments were conducted with male and female C57BL/6 (Charles River Laboratory) mice, 10–20 weeks old at the start of behavioral training. They were first prepared for head fixation under general isoflurane anaesthesia (1.5–2%) as previously described[2]. Briefly, a custom-made titanium head bar was fixed on the skull with a cyanoacrylate adhesive (ergo 5011, IBZ Industrie) and dental cement to allow for subsequent head fixation. They were housed in an animal facility, maintained on a 12:12 light/dark cycle (temperature between 21 and 22 °C, humidity between 55 and 65%) and were placed under a water restriction regime (1 ml/day) 1 week before the start of experiments. The experiments were performed during the light phase of the cycle. The animals did not undergo any previous surgery, drug administration or experiments and were housed in groups of maximum five animals per cage. All procedures complied with and were approved by the Institutional Animal Care and Use Committee of the University of Geneva and Geneva veterinary offices.

**Human participants**. The cohort included 24 participants aged between 21 and 48 years (mean ± s.d. = 29.25 ± 7.89, 12 females) with no history of somatosensory injury or disease, no psychiatric disorder and no substance abuse. Prior to study participation, all gave informed consent and received a 20 CHF/h monetary retribution at their last session. All experimental procedures approved by the ethics

commission of the Geneva canton (CCER, Cantonal Swiss Ethics Committee on research involving humans, University of Geneva).

**Vibrotactile stimulation**. Vibrotactile stimulation was delivered with piezoelectric stack actuators (P-841.3 for mouse and P-841K191 for human experiments, Physik Instrumente). The stimulation endpoint was a metal rod (2 mm diameter) mounted either vertically (for human fingertip stimulation) or horizontally (for mouse forepaw stimulation) on the actuator with an M3 screw. Actuator position was monitored with a strain gauge sensor and the actuator and sensor controllers (E-504 and E-509.S3 for mouse, E-618.1 and E-509.S1 for human experiments, Physik Instrumente) operated either in closed loop (450 Hz center frequency experiment in mice) or open loop (all other experiments) modes. Operating in open loop mode was necessary in order to produce the full range of frequencies tested in the study. The recorded sensor signals were analyzed offline in temporal and spectral domains and revealed that open loop operation did not compromise the integrity of the vibratory stimuli. The stimuli were pure sinusoids (250 or 500 ms duration, 25 or 50 ms linear onset/offset ramps) sampled at either 10, 20, or 30 kHz (USB-6353, National Instruments). Although naturally occurring vibrations are nonstationary and typically have a broad spectrum, pure sinusoidal stimuli can be used to better quantify perceptual responses. The amplitude of the sinusoids was calibrated based on sensor measurements in order to produce the required actuator displacements. Recalibration was performed regularly to guarantee stimulus consistency over time.

**Behavioral procedures**. Mouse behavior was controlled with real-time routines running on Linux (BControl, brodylab.princeton.edu/bcontrol) and interfaced with Matlab (Mathworks) running on a separate PC. Human behavior was controlled with custom routines programmed in Matlab.

*Frequency discrimination task in mice*. We used a go/no-go task to train mice to discriminate frequencies of vibrotactile stimuli with their forepaw. They were head-fixed and positioned inside a tube (25 mm inner diameter) such that their right forepaw held the stimulator to maintain balance, while their left forelimb was blocked from protruding outside the tube. The trial started with a 1 s period requiring continuous holding of the stimulator followed by stimulus delivery. The hold interval was reset upon every paw release. A white noise sound was played over loud speakers at the moment of vibratory stimulation, thereby acting simultaneously as an auditory mask and as a stimulus cue. Following stimulus presentation, mice had to initiate licking of a water spout for Go frequencies and refrain from licking for No-go frequencies, within a 2 s period. Hit trials (licking for go-stimuli) were rewarded by a drop of water, misses (no licking for go-stimuli) and false alarms (licking for no-go stimuli) were punished by a 1–6 s timeout. Correct rejections (no licking for no-go stimuli) were neither rewarded nor punished. A new trial was initiated after licking ceased for a minimum of 2 s. To minimize licking response bias, one of two strategies was used. In the first, a minimum of three consecutive correct rejection responses were required before a go trial was presented. In the second, the probability of a go trial ($P_{go}$) was determined according to the double sigmoidal model:

$$P_{go}(\text{bias}) = 1 - \frac{0.5}{1 + \left(\frac{\text{bias}+1}{\tau_1 - 1}\right)^{S_1}} - \frac{0.5}{1 + \left(\frac{\text{bias}+1}{\tau_2 - 1}\right)^{S_2}} \qquad (1)$$

Where S1 and S2 are the slopes at the chosen inflection points $\tau 1 = -0.5$ and $\tau 2 = 0.5$, respectively. The steepness of the slopes was arbitrarily chosen to be S1 = 16 and S2 = 44. The bias value was defined as the difference in the fraction of correct responses between no-go and go trials in the last 20 trials.

We tested three different frequency ranges with three groups of mice: a low range (four mice; center frequency: 450 Hz; no-go stimuli: 310, 345, 380, and 415 Hz; go-stimuli: 485, 520, 555, and 590 Hz), a middle range (four mice; center frequency: 1000 Hz; no-go stimuli: 900, 925, 950, and 975 Hz; go-stimuli: 1025, 1050, 1075, and 1100 Hz), and a high range (six mice; center frequency: 1600 Hz; no-go stimuli: 1500, 1525, 1550, and 1575 Hz; go-stimuli: 1625, 1650, 1675, and 1700 Hz). The four mice tested on the middle range were also part of the high range group. The two ranges were tested more than two weeks apart. In the first 7–14 sessions, the mice performed the task at a fixed reference amplitude (5.6 μm for the low range and 3.8 μm for the middle and high ranges). In the last six sessions (five for the low range), nontimed probe amplitudes were introduced in 30% of the trials to test the effect of amplitude change on frequency discrimination. The probe trials occurred pseudo-randomly and followed the same go/no-go rules as the 70% of trials delivered at the trained reference amplitude. A single probe amplitude was tested in each session (8.6, 7.1, 4.1, 2.6, or 1.1 μm for the low range; 6.9, 5.4, 4.8, 2.7, 2.1, or 1.8 μm for the middle and high ranges). Each stimulus frequency-amplitude pair was repeated at least 10 times in a single session.

*Frequency discrimination task in humans*. We used a two-alternative forced choice task to test vibrotactile pitch perception in 9 healthy human participants (age mean ± s.d. = 27.56 ± 5.03; 5 females). An additional six subjects performed the task but were excluded from the analysis after realizing the actuator failed to generate vibrations at one of the amplitudes due to a coding error. Participants sat comfortably in a dark room and positioned their right forearm on a vibration

isolation table. The stimulator endpoint (a punctuate probe of 3 mm diameter mounted on the piezo stack) was placed in contact with the fingertip of their index finger, with their hand either in the palm down (four subjects) or palm up (five subjects) position. We did not control for the contact force as it was previously reported to play no role in behavioral performance[8]. The participants wore noise canceling headphones (3 M Peltor WorkTunes Pro HRXS220A) and masking white noise was played throughout the session. The task was guided with visual cues displayed on a 60 inch monitor viewed at a 140 cm distance. Each trial started with a 0.5 s pre-stimulus interval during which a red fixation dot was displayed, followed by a 2.5 s stimulus interval cued with the fixation dot turning green. During this interval, two successive vibrations (0.5 s duration each) were delivered to the fingertip, preceded, separated and followed by a 0.5 s silent period. A nontimed answer period followed in which the words 'First' and 'Second' appeared on the screen. The participants were instructed to select whether the first or second vibration had a higher frequency with the push of a button (Stream Deck Mini) held in their left hand. The instruction was to focus on the frequency and ignore the amplitude; the two terms were clearly explained to the participants prior to experiment start. One of the two stimuli (the standard) was always at the same reference frequency $f_{REF}$ and the other at a changing test frequency $f_{TEST} = f_{REF} \pm \Delta$. The order of the standard and test was randomized, but the comparison of the test relative to the standard was measured during analysis. The amplitude of the test stimuli was kept constant at $A_{TEST} = 11.8$ μm and the amplitude of the standard was consistently changed between seven different values $A_{REF} = 7.4, 8.4, 9.8, 11.8, 14.2, 16.5,$ and $18.9$ μm. Before the start of each session, participants received training trials with $\Delta = \Delta_{MAX}$, repeated until they performed 10 correct answers in a row for each $A_{REF}$. During the training trials, feedback about correct performance was given by highlighting in green (for correct) or red (incorrect) the selected response. The purpose of these training trials was to ensure that the subjects understood the instructions and were repeated until they performed close to 100% correct for the easiest comparisons (i.e., $f_{TEST} = f_{REF} \pm \Delta_{MAX}$). No feedback was given on the subsequent test trials. The test stimuli were presented using a custom staircase adaptive procedure. For each $A_{REF}$, test stimuli started with $\Delta = \Delta_{MAX}$. After each correct or incorrect answer, $\Delta$ was lowered or increased by d$\Delta$ (its rate of change), respectively. After three successive changes in the same direction, d$\Delta$ was doubled and after each change direction reversal, d$\Delta$ was halved. These adjustments were made independently for $f_{TEST} > f_{REF}$ and $f_{TEST} < f_{REF}$. Each participant repeated the experiment five times, each time with a different $f_{REF}$, in separate sessions. The five tested $f_{REF}$ were 160, 200, 280, 440, and 480 Hz. Their respective $\Delta_{MAX}$ were 64, 128, 128, 256, and 256 Hz, their respective minimum rates of change d$\Delta$ were 8, 8, 16, 32, and 32 Hz, and their respective maximum rates of change d$\Delta$ were 16, 32, 32, 64, and 64 Hz. In each session, the $A_{REF}$ values were randomly sampled without replacement and a minimum of 500 trials were performed (i.e., at least 70 at each $A_{REF}$). The participants were given an option to take a break after every block of ten trials.

*Detection task in mice*. In order to determine their perceptual thresholds, we used a two-alternative forced choice task to train seven mice in a vibrotactile detection task. Mice were trained to lick, in the response period, toward either a right or left reward spout if a vibrotactile stimulus was present or absent during the preceding stimulus period, respectively. All other experimental conditions were as described above in the frequency discrimination task. Correct responses were rewarded with a drop of water at the corresponding spout and incorrect responses were not punished by a timeout. Trials without a response were neither rewarded nor punished and occurred on <5% of trials. To minimize a direction bias, the trial type was chosen pseudo-randomly by allowing a maximum of two trials of the same type in a row (50% chance of occurrence for each otherwise). We tested the perceptual thresholds at seven different frequencies (200, 450, 700, 1000, 1300, 1600, and 2000 Hz) in separate sessions and in randomized order. Between 1 and 3 sessions were tested in a single day and the same session (i.e., frequency) was repeated up to five times on separate days per mouse. Prior to testing, the mice were first trained on all frequencies at the largest possible amplitude that the actuator could produce at each frequency. This value ranged from 10 μm (at 200 Hz) to 1 μm (at 2000 Hz). The training lasted 10 days, followed by a 2-month break (COVID-19) and a second training period of 10–12 days. Testing of each frequency started at the largest possible amplitude and was progressively attenuated in −4 dB steps after every six vibration trials (total of ≈12 trials) if the proportion of correct responses exceeded 70%. The amplitude was increased by 4 dB if the proportion of correct responses decreased below 60% after every ≈12 trials (including at least six vibration trials). To determine the perceptual threshold at each frequency, we compared the ratio of correct responses for each bout of trials at a given amplitude to chance (i.e., 0.5) using the one-sided binomial test. The threshold was the lowest amplitude of the session for which the test yielded a significance level of <0.05. The thresholds of repeated sessions were averaged and allowed establishing the V-shaped vibrotactile sensitivity curves (Fig. 2C).

*Detection task in humans*. We used a two-alternative forced choice task to determine the perceptual thresholds across a wide range of vibration frequencies in 19 healthy human participants (age mean ± s.d. = 30.21 ± 8.38, 9 females). Each trial started with a 0.5 s pre-stimulus interval (red fixation dot) followed by a 3.25 s stimulus interval. The stimulus interval consisted of two successive 1.5 s active

periods (cued by green dots on the display) separated by a 0.25 s passive period (red dot on display). A 0.5 s vibratory stimulus was delivered at a random time either during the first or the second active period. The participants were instructed to answer in which of the two periods the stimulus was present by either selecting 'First' or 'Second' on the display with the push of a button. We tested 14 different vibration frequencies (10, 25, 50, 75, 100, 200, 300, 400, 500, 600, 700, 800, 900, and 1000 Hz) in separate blocks and in randomized order. To determine the perceptual threshold for each, we used a 3-down, 1-up adaptive staircase procedure. For each frequency, the vibration amplitude started at its maximal value (i.e., the maximal travel range of the piezo stack at that frequency) and was decreased by Δ dB after three successive correct answers and increased by Δ dB after one incorrect answer. Δ started at 12 dB and was halved after each direction reversal, but maintained at a minimum of 3 dB. The testing stopped after five direction reversals and the detection threshold was taken as the mean amplitude of the last 10 trials. All other experimental conditions were as described above in the frequency discrimination task.

*Amplitude discrimination task in humans.* We used a two-alternative forced choice task to test vibrotactile intensity perception in 10 healthy human participants (age mean ± s.d. = 27.20 ± 5.39; 5 females). All experimental details were as described above for the frequency discrimination task. The participants were instructed to select whether the first or second vibration had higher amplitude. The instruction was to focus on the amplitude and ignore the frequency. One of the two stimuli (the standard) was always at the same reference amplitude $A_{REF}$ and the other at a changing test amplitude $A_{TEST} = A_{REF} ± Δ$. The frequency of the test stimuli was kept constant at $f_{TEST} = 200$ Hz and the frequency of the standard was consistently changed between seven different values $f_{REF} = 75, 100, 150, 200, 266, 400,$ and 534 Hz. Each participant repeated the experiment four times, each time with a different $A_{REF}$, in separate sessions. The task structure and testing procedure were analogous to those described above in the frequency discrimination task. The four tested $A_{REF}$ were 6, 8, 10, and 12 µm. Their respective $Δ_{MAX}$ were different for $A_{TEST} > A_{REF}$ than for $A_{TEST} < A_{REF}$ due to the amplitude limitations imposed by the hardware. For $A_{TEST} > A_{REF}$, the respective $Δ_{MAX}$ were 12, 10, 8, and 6 µm, and for $A_{TEST} < A_{REF}$, the $Δ_{MAX}$ were 5, 7, 9, and 11 µm. The minimum rate of change dΔ was 1 µm, and the maximum rate of change was 3 µm for all $A_{REF}$.

*Discrimination task along equal-intensity curves.* We used a two-alternative forced choice task to control for the possible influence of perceived intensity on pitch perception in six healthy human participants (age mean ± s.d. = 31.17 ± 5.38; 3 females). All experimental details were as described above for the frequency discrimination task. The participants were instructed to select whether the first or second vibration had higher frequency. One of the two stimuli (the standard) was always at the same 440 Hz reference frequency and the other at a changing test frequency uniformly distributed between 352 and 528 Hz. The amplitude of each stimulus was different: it was obtained from the cubic-spline interpolated equal-intensity data as depicted in Supplementary Fig. 1. An initial training phase lasted between 80 and 250 trials (depending on the participant's performance) during which feedback about correct answers was given (red or green highlight of the answer after each response). A subsequent test phase lasted between 500 and 600 trials during which no feedback was given. In the test phase, two probe standard amplitudes were introduced on 1/3 of the trials. The experiment was repeated twice; once for the 10 µm and once for the 12 µm equal-intensity curve. The amplitude change factors (of the test relative to the standard) of the two probe amplitudes were 1.67 and 0.77 for the 10 µm curve, and 1.25 and 0.85 for the 12 µm curve.

#### Data analysis
*Psychometric curve fitting.* In the frequency discrimination tasks, we analyzed the fraction of lick responses in mice and the fraction of test stimuli reported to be higher relative to the standard in humans, as a function of vibration frequency. The data were fit with a sigmoid function (i.e., a cumulative Gaussian) assuming equal asymptotes, using the psignfit Matlab toolbox[23]. Only for the middle range data in mice could we not assume equal asymptotes and therefore fitted in addition the lapse rate and guess rate parameters.

*Pitch perception fitting.* We identified that pitch perception can be expressed as $A^k × f$ by fitting the frequency shift ratio $μ/μ_{REF}$ as a function of amplitude change factor $A/A_{REF}$ as:

$$\frac{μ}{μ_{REF}} = \left(\frac{A_{REF}}{A}\right)^k \qquad (2)$$

Where $μ$ and $μ_{REF}$ are the mean parameters of the psychometric curve fits to the behavioral responses obtained for the probe/test amplitudes $A$ and the reference amplitude $A_{REF}$, respectively. The fitted parameter k was the one minimizing the sum of squared residuals between measured and predicted values using the regress function in Matlab. Accordingly, all amplitude $A$ and frequency $f$ pairs yielding the same $A^k × f$ value (the one equal to $A_{REF}^k × f_{REF}$) evoke the same pitch percept. Note that in the human experiments, even though $A_{REF}$ was varied and $A_{TEST}$ was kept constant, we still use the $A/A_{REF}$ ratio for fitting the k parameter. The equal-

pitch curves in Fig. 1C,F and Fig. 2A, B were plotted by multiplying the frequency shift ratio values by the center/reference frequency and the amplitude change factor values by the reference amplitude.

*Intensity perception fitting.* The equal-intensity datapoints (Fig. 4) were fit with a sum of exponential function:

$$A = a*e^{bf} + c*e^{df} \qquad (3)$$

Where $A$ is amplitude, $f$ is frequency and $a, b, c, d$ are the free parameters. Because the equal-intensity datapoints (Fig. 4) were only evaluated within a limited frequency range (up to 534 Hz) the fits are for descriptive purposes only. The fitted parameters are likely not representative of the full extent of the equal-pitch datapoints.

*Significant responses.* The $A^k × f$ fit was deemed significant when the 95% confidence intervals of the fitted k parameter did not include zero. For non-significant fits, k was made equal to zero.

*Statistics.* Data were taken from distinct subjects and no subject was measured repeatedly. No statistical methods were used to predetermine sample size. No randomization was required as our study did not involve separating subjects into control and experimental groups. Analyses of data comparing different experimental conditions in the same subjects were performed by blinded researchers. All data analyses were performed with custom written routines in Matlab (Mathworks).

**Reporting summary.** Further information on research design is available in the Nature Research Reporting Summary linked to this article.

### Data availability
The data generated in this study have been deposited in the YARETA database under accession code [https://doi.org/10.26037/yareta:p73yr7n7bnfnbp6wrt355g5q5e]. Source data are provided with this paper.

### Code availability
The Matlab code for generating and analyzing data in the current study has been deposited in the YARETA database under the accession code [https://doi.org/10.26037/yareta:p73yr7n7bnfnbp6wrt355g5q5e].

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

## Author contributions

M.P. and D.H. designed the study. M.P., D.K., A.N., and K.S.L. performed the experiments. M.P. and D.K. analyzed the results. M.P., D.K., and D.H. interpreted the results and wrote the paper.

## Competing interests

The authors declare no competing interests.
