## [Peer Review File · Nature Communications]

A common computational principle for vibrotactile pitch perception in mouse and humanREVIEWER COMMENTS

Reviewer #2 (Remarks to the Author):

This is a follow-up manuscript to the 2019 paper by Prsa and Huber that identifies the trade-off of stimulus frequency versus amplitude in the perception of a mechanical vibration. The authors show that the original form of scaling applies to humans, albeit with differences in the offsets and slopes of their curves, and they further expand on their mouse data.

I enjoyed the reading the paper. I have little to comment on except that it would be useful to have a parameterization of the data in Figures 2C and 3A with regard to the frequency for minimum amplitude, which also defines the symmetry axis (e.g., $f_0 = 1000$ Hz in Fig 2A). This might give theorists or modelers some insight - at the least a target function. Something like:

$$a=0.5; b=0.4; A=\exp((\text{abs}(f-f_0))^{(a+b)})/(\text{abs}(f/f_0)^a);$$

In any case, this is a minor point and the manuscript can be published as is.

Reviewer #3 (Remarks to the Author):

This manuscript makes an important contribution to our understanding of tactile pitch perception. That is, that there is an interaction between stimulus intensity and frequency in determining the perceived tactile pitch in mice and humans. Furthermore, the manuscript demonstrates how this interaction can be explained in terms of sensitivity thresholds for different frequencies of tactile vibration, and they show that this explanation generalizes across mice and humans.

The experiments use simple but elegant psychophysical methods. They address an important question and offer valuable new insights into tactile perception.

Major concerns:

1. Replication of previous results. The manuscript could easily be re-written starting with Fig 2 A, and then lead to the human data (Fig 1B and 2B,C). This would change some of the author's storytelling at the beginning of the results, but would avoid them replicating their previously published data and results (Fig 1A-C). For me, it would be preferable to avoid replicating these previously published findings as it is not required here. Start by saying "here is this interesting relationship between frequency and amplitude in mice" (Fig 2A), and then "it also happens in humans" (Fig 1D-F and 2B), and finally "it all makes sense across species in terms of sensitivity at different frequencies" (Fig 2C - which would work better as an independent figure, in my opinion). And then end with the current final figure.

2. Statistics. The equal pitch curve fits (e.g. Fig 1F) all look convincing to me. However, I'm not as convinced by the cubic spline interpolations of the equal intensity curves (Fig 3A). These look overfitted in places (e.g. largest frequency value). Have the authors tried fitting a simpler curve with fewer parameters or a lower order? Can they statistically test their fits against these simpler alternatives? Or is there a previous literature justifying the use of the cubic spline for this dataset?

3. Discussion of auditory pitch. The discussion of auditory pitch perception in the abstract, introduction and discussion contains misconceptions that will not be well received by experts in auditory neuroscience. For example, in contrast to the wording of the abstract and Intro line 32, pitch is "a" main perceptual characteristic of sound, but not necessarily "the" main one. Many important sounds do not evoke a pitch percept (e.g. footsteps, rustling leaves, whispering). Many auditory neuroscientists might therefore argue that timbre is "the" most important perceptual dimension of sound. Cochlear implant users have terribly poor pitch perception, yet function and communicate in the world quite well. This is a small but crucial distinction. As another example, the parallels drawn between auditory and tactile pitch in the Discussion are based on speculations and oversimplifications of a much more complex auditory pitch mechanism, and are only relevant for the pitch of pure tones, which rarely occur in the natural world (see my points 8 and 9 under minor comments, below).

Furthermore, auditory pitch is beyond the scope of the current experiments, which all examine tactile perception only. I would strongly suggest that the abstract should not discuss auditory pitch at all, as it is a complex topic that it is difficult to cover briefly. The sections on auditory pitch in

the introduction and discussion should either be omitted, or corrected and shortened.

Minor comments:

1. Abstract: delete "actual"
2. Abstract: "equal pitch perception" is an unclear choice of terminology. In addition, pitch is multidimensional, including pitch height, scale, salience. Is this also true in vibro-tactile pitch? If so, is the pitch perceived equivalent in all these dimensions? To be more precise, describe what you objectively measured instead of inferring the percept.
3. It seems to me that the evidence cited in lines 40-41 are very important to understanding the background and current questions in this research field. Therefore, I'd like to have the results of these two studies described, so that we can better understand what the open research questions are more specifically. If the authors need more space to fit it in, they could shorten or omit the discussion of auditory pitch perception in the preceding paragraph, which is not directly relevant to this manuscript.
4. Line 43: remove "ideally". This is a fine set of experiments, but not necessarily the ideal experiments! Also, change the wording so this changes from a hypothetical suggestion to a statement about what you did here.
5. "vibrotactile" is hyphenated in some places, and not in others.
6. Line 73: Explain why were different frequency ranges used for mice (> 450Hz) and humans (<440Hz) in Fig 2. This would seem to introduce a confound in comparing the results across species. The relationship to amplitude sensitivity is not obvious at this point in the paper.
7. Line 104: It would be helpful for the non-expert to describe the differences between mouse and human Pacinian corpuscle distribution that are relevant here.
8. In the final paragraph of the Discussion, the type of auditory pitch being discussed here is the pitch of pure tones, which is a small subset of pitch-evoking sounds and the simplest type possible. This is worth pointing out because this simple "frequency" perception becomes much more complicated for the pitch of real-world, complex sounds. Complex pitch is what one normally thinks about when discussing auditory pitch, and the elegant explanation of auditory frequency perception given here would not hold for the pitch of complex sounds.
9. Point 8 above is highly relevant to the claim on line 135 about a "common mechanism" for pitch in audition and somatosensation. The mechanisms for complex pitch perception are outside of the simple pure tone pitch perception discussed here, and may very well differ from those that exist for touch. Therefore, you must make it very clear when you talk about auditory pitch in your manuscript that you are talking about mechanisms for perceiving the pitch of a pure tone only. There is too much speculation in this final paragraph on the parallels between auditory and tactile pitch given that only the latter are examined in this study. This type of interesting speculation might better fit in a review article on the topic, or at least in the Discussion section of a manuscript that investigates both sensory modalities.

Reviewer #4 (Remarks to the Author):

Prsa et al study the impact of vibration amplitude on vibro-tactile pitch in mice and humans. They find an interaction, as described earlier, but at the same time find that humans can -after training- distinguish amplitudes and frequencies well. I find that the authors have not excluded that insufficient separation of amplitude and frequency cues during training may underlie the observed power law interaction effects between the two. The paper therefore falls short in convincingly showing a new computational principle (l.20).

Major point:

In Figure 1, the mice are trained at different frequencies with the same vibration amplitude, but these will be perceived by them as having different intensity. So it is no surprise that they will be using intensity as an additional cue apart from frequency, and it is incorrect to call the curves in Fig. 1C,F, 2A,B equal pitch curves (l.107) or to say that two stimuli differing in both amplitude and frequency are perceptually indistinguishable (l.19) if this has not been tested more systematically. If the training were done with stimuli that were perceived as having equal intensity (i.e. along the mouse equivalent of one of the green curves in Fig. 3B), I would expect that the described interaction effect would largely disappear. The shape of the curves in Fig. 1C,F, 2A,B provides some information about how intensity and frequency are weighted, but not in a very systematic

way, as their shape should also depend on the training and test conditions.

The human experiments described in Figure 1 were also not designed to stringently separate the intensity and frequency cues. During the instructions the difference between the two was verbally explained, but not systematically trained. Only easy trials differing strongly in frequency (and also in amplitude) were being trained. It is therefore no wonder that the intensity cues are not entirely disregarded, whereas Fig. 3 shows clearly that this cue can be perceived well, independently from the frequency cue.

Minor points:

I.32: "the main property of airborne vibrations (i.e. sound) is pitch perception" -> "the main distinguishing property of airborne vibrations (i.e. sound) is its pitch"

I.42: "to" missing.

I.99 ostensibly

I.172: the shape of the reference curve in 1B depends on the test frequency, which suggests a Bayesian strategy of the mice: as the test stimuli get more difficult, the false-positive rate of the ref stimuli seems to go up as well.

I.252: The methods for the mouse tasks are somewhat unusual with a categorization task that is also a detection task (Fig. 1B, ACF=0.2), and which is implemented as a Go/NoGo task, whereas the detection threshold task is implemented as a two-alternative task.

I.258: the authors should check the equation for calculating Pgo. It seems to involve negative numbers raised to a non-integer power.

I.310: Why a staircase procedure to obtain a psychometric curve?

I.321: Why is this called a forced choice task if trials without a response were neither rewarded nor punished?

Discussion: Stevens' rule for the pitch shift is a small effect (about a semitone max, see the Cohen ref), and it has not always been easily reproduced. I'm not sure the superficial resemblances with the experiments in the present paper justify calling these a fundamental relationship between spectral sensitivity and pitch perception (I.22).

REVIEWER COMMENTS

Reviewer #2 (Remarks to the Author):

This is a follow-up manuscript to the 2019 paper by Prsa and Huber that identifies the trade-off of stimulus frequency versus amplitude in the perception of a mechanical vibration. The authors show that the original form of scaling applies to humans, albeit with differences in the offsets and slopes of their curves, and they further expand on their mouse data.

I enjoyed the reading the paper. I have little to comment on except that it would be useful to have a parameterization of the data in Figures 2C and 3A with regard to the frequency for minimum amplitude, which also defines the symmetry axis (e.g., $f_0 = 1000$ Hz in Fig 2A). This might give theorists or modelers some insight - at the least a target function. Something like:

$$a=0.5; b=0.4; A=\exp((\text{abs}(f-f_0)^{(a+b)})/(\text{abs}(f/f_0)^a));$$

In any case, this is a minor point and the manuscript can be published as is.

Answer:

We thank the reviewer for the suggestion to parametrize the amplitude threshold curves and equal intensity curves (new Fig. 3 and 4). After evaluation of several models, we found that a sum of exponentials function provides the best fit. We added the fitted curves in the figures and their description in the methods (p. 14, lines 1-7). It should however be noted that the equal-intensity data points (new Fig. 4) could only be evaluated within a limited frequency range (up to 534 Hz only) due to hardware limitations. The fits to this incomplete data set is therefore for descriptive purposes only. The actual fitted parameters are likely not representative of the full extent of the equal-pitch data points.

We also found that the same fit captured well the human threshold data but not the mouse threshold data (new Fig. 3), especially the 1 kHz trough. We therefore prefer not to include the fit for the threshold data.

Reviewer #3 (Remarks to the Author):

This manuscript makes an important contribution to our understanding of tactile pitch perception. That is, that there is an interaction between stimulus intensity and frequency in determining the perceived tactile pitch in mice and humans. Furthermore, the manuscript demonstrates how this interaction can be explained in terms of sensitivity thresholds for different frequencies of tactile vibration, and they show that this explanation generalizes across mice and humans.

The experiments use simple but elegant psychophysical methods. They address an important question and offer valuable new insights into tactile perception.

Major concerns:

1. Replication of previous results. The manuscript could easily be re-written starting with Fig 2 A, and then lead to the human data (Fig 1B and 2B,C). This would change some of the author's storytelling at the beginning of the results, but would avoid them replicating their previously published data and results (Fig 1A-C). For me, it would be preferable to avoid replicating these previously published findings as it is not required here. Start by saying "here is this interesting relationship between frequency and amplitude in mice" (Fig 2A), and then "it also happens in humans" (Fig 1D-F and 2B), and

finally "it all makes sense across species in terms of sensitivity at different frequencies" (Fig 2C - which would work better as an independent figure, in my opinion). And then end with the current final figure.

Answer:

We thank the reviewer for this suggestion. We have reorganized the results section accordingly:

- First paragraph and Fig. 1: pitch perception as a function of frequency and amplitude in mice.
- Second paragraph and Fig. 2: pitch perception as a function of frequency and amplitude in humans.
- Third paragraph and Fig. 3: relationship between pitch perception and sensitivity.
- Fourth paragraph and Fig. 4: distinction between equal pitch and intensity curves.

2. Statistics. The equal pitch curve fits (e.g. Fig 1F) all look convincing to me. However, I'm not as convinced by the cubic spline interpolations of the equal intensity curves (Fig 3A). These look overfitted in places (e.g. largest frequency value). Have the authors tried fitting a simpler curve with fewer parameters or a lower order? Can they statistically test their fits against these simpler alternatives? Or is there a previous literature justifying the use of the cubic spline for this dataset?

Answer:

We now provide a parametric fit to the equal-pitch curves in the new Fig. 4 (see Methods p. 14, lines 1-7). It should however be noted that the equal-intensity data points (new Fig. 4) could only be evaluated within a limited frequency range (up to 534 Hz only) due to hardware limitations. The fits to this incomplete data set is therefore for descriptive purposes only. The actual fitted parameters are likely not representative of the full extent of the equal-pitch data points. We also found that the same fit captured well the human threshold data but not the mouse threshold data (new Fig. 3), especially the 1 kHz trough. We therefore prefer not to include the fit for the threshold data.

3. Discussion of auditory pitch. The discussion of auditory pitch perception in the abstract, introduction and discussion contains misconceptions that will not be well received by experts in auditory neuroscience. For example, in contrast to the wording of the abstract and Intro line 32, pitch is "a" main perceptual characteristic of sound, but not necessarily "the" main one. Many important sounds do not evoke a pitch percept (e.g. footsteps, rustling leaves, whispering). Many auditory neuroscientists might therefore argue that timbre is "the" most important perceptual dimension of sound. Cochlear implant users have terribly poor pitch perception, yet function and communicate in the world quite well. This is a small but crucial distinction. As another example, the parallels drawn between auditory and tactile pitch in the Discussion are based on speculations and oversimplifications of a much more complex auditory pitch mechanism, and are only relevant for the pitch of pure tones, which rarely occur in the natural world (see my points 8 and 9 under minor comments, below).

Furthermore, auditory pitch is beyond the scope of the current experiments, which all examine tactile perception only. I would strongly suggest that the abstract should not discuss auditory pitch at all, as it is a complex topic that it is difficult to cover briefly. The sections on auditory pitch in the introduction and discussion should either be omitted, or corrected and shortened.

Answer:

We thank the reviewer for these very insightful comments.

In the abstract, we have removed the reference to auditory pitch and focus exclusively on vibrotactile perception. In the introduction, we have rewritten the statement in question in a more nuanced way:

"In the auditory system, airborne vibrations (i.e. sound) can evoke pitch perception ..."

In the discussion, we still think that it is relevant to speculate about the parallels between auditory pitch and vibrotactile pitch perception. We have however toned down the description of Steven's

psychoacoustic effect and focus more on the neurophysiological findings about how neuronal tuning curves in cortex and the location of maximum cochlear excitation shifts with sound intensity. We still speculate about the parallels between auditory and somatosensory processing (as has been often done in the literature), but mainly in terms of neural representation.

Minor comments:

1. Abstract: delete "actual"

Corrected.

2. Abstract: "equal pitch perception" is an unclear choice of terminology. In addition, pitch is multidimensional, including pitch height, scale, salience. Is this also true in vibro-tactile pitch? If so, is the pitch perceived equivalent in all these dimensions? To be more precise, describe what you objectively measured instead of inferring the percept.

We now define in the abstract the "pitch perception" terminology as "the perception of how high or low a vibration's frequency is", which corresponds to how it has been previously defined in the literature (Morley and Rowe, 1990).

3. It seems to me that the evidence cited in lines 40-41 are very important to understanding the background and current questions in this research field. Therefore, I'd like to have the results of these two studies described, so that we can better understand what the open research questions are more specifically. If the authors need more space to fit it in, they could shorten or omit the discussion of auditory pitch perception in the preceding paragraph, which is not directly relevant to this manuscript.

We now explicitly describe the finding of the Morley and Rowe study in the introduction:

"human subjects can perceive that a vibration has higher or lower pitch than a frequency-matched comparison if its amplitude is changed (Morley and Rowe, 1990)"

We do not explicitly describe in the introduction the findings of our previous publication (Prsa et al, 2019) as it is essentially the same result as the leftmost panel in Fig. 1C.

4. Line 43: remove "ideally". This is a fine set of experiments, but not necessarily the ideal experiments! Also, change the wording so this changes from a hypothetical suggestion to a statement about what you did here.

We thank the reviewer for this suggestion and have rewritten the sentence accordingly.

5. "vibrotactile" is hyphenated in some places, and not in others.

Corrected

6. Line 73: Explain why were different frequency ranges used for mice (> 450Hz) and humans (<440Hz) in Fig 2. This would seem to introduce a confound in comparing the results across species. The relationship to amplitude sensitivity is not obvious at this point in the paper.

After having obtained the opposite effect of amplitude changes on pitch perception in humans and mice at 440/450 Hz, we started to speculate that this is due to their different perceptual spectral ranges, hence the choice for these frequencies. Therefore, these choices can only be justified retrospectively.

7. Line 104: It would be helpful for the non-expert to describe the differences between mouse and human Pacinian corpuscle distribution that are relevant here.

We now provide a detailed description of the difference in Pacinian corpuscle distribution between mice and primates in the first sentence of the discussion.

8. In the final paragraph of the Discussion, the type of auditory pitch being discussed here is the pitch of pure tones, which is a small subset of pitch-evoking sounds and the simplest type possible. This is worth pointing out because this simple "frequency" perception becomes much more complicated for the pitch of real-world, complex sounds. Complex pitch is what one normally thinks about when discussing auditory pitch, and the elegant explanation of auditory frequency perception given here would not hold for the pitch of complex sounds.

Based on the reviewer's previous comment, we have removed the discussion of auditory pitch perception from this section. When discussing cortical auditory responses, we have now specified that the responses relate to pure tones.

9. Point 8 above is highly relevant to the claim on line 135 about a "common mechanism" for pitch in audition and somatosensation. The mechanisms for complex pitch perception are outside of the simple pure tone pitch perception discussed here, and may very well differ from those that exist for touch. Therefore, you must make it very clear when you talk about auditory pitch in your manuscript that you are talking about mechanisms for perceiving the pitch of a pure tone only. There is too much speculation in this final paragraph on the parallels between auditory and tactile pitch given that only the latter are examined in this study. This type of interesting speculation might better fit in a review article on the topic, or at least in the Discussion section of a manuscript that investigates both sensory modalities.

When speculating about the common principles between the two systems in the discussion, we now explicitly state that we refer to pure tones and sinusoidal vibrations. We still think that this speculation is relevant given the similar neural encoding principles and would like to keep it in the manuscript.

Reviewer #4 (Remarks to the Author):

Prsa et al study the impact of vibration amplitude on vibro-tactile pitch in mice and humans. They find an interaction, as described earlier, but at the same time find that humans can -after training- distinguish amplitudes and frequencies well. I find that the authors have not excluded that insufficient separation of amplitude and frequency cues during training may underlie the observed power law interaction effects between the two. The paper therefore falls short in convincingly showing a new computational principle (l.20).

Major point:

In Figure 1, the mice are trained at different frequencies with the same vibration amplitude, but these will be perceived by them as having different intensity. So it is no surprise that they will be using intensity as an additional cue apart from frequency, and it is incorrect to call the curves in Fig. 1C,F, 2A,B equal pitch curves (l.107) or to say that two stimuli differing in both amplitude and frequency are perceptually indistinguishable (l.19) if this has not been tested more systematically. If the training were done with stimuli that were perceived as having equal intensity (i.e. along the mouse equivalent of one of the green curves in Fig. 3B), I would expect that the described interaction effect would largely disappear. The shape of the curves in Fig. 1C,F, 2A,B provides some information about how intensity and frequency are weighted, but not in a very systematic way, as their shape should also depend on the training and test conditions.

The human experiments described in Figure 1 were also not designed to stringently separate the intensity and frequency cues. During the instructions the difference between the two was verbally explained, but not systematically trained. Only easy trials differing strongly in frequency (and also in amplitude) were being trained. It is therefore no wonder that the intensity cues are not entirely disregarded, whereas Fig. 3 shows clearly that this cue can be perceived well, independently from the frequency cue.

Answer:

We thank the reviewer for this very important insight. Indeed, theoretically, frequency discrimination might have been guided by intensity perception. To explicitly show that the observed psychometric shifts are not guided by differences in perceived intensity between the tested stimuli, we conducted an additional control experiment. As the reviewer suggested, if the subjects were trained to perform the discrimination along one of the equal-intensity green curves (Fig. 3B) we can expect one of the following two outcomes:

-Probe amplitudes should not evoke any shifts in psychometric curves if pitch perception is not a function of amplitude.

-Probe amplitudes should still evoke shifts if pitch perception is a function of both frequency and amplitudes. These shifts could however not be interpreted in any meaningful manner.

We carried out the following experiment:

We took the 10 um equal intensity curve in Fig. 4A and chose 8 test stimuli that lie on that curve equally spaced between 352 Hz and 528 Hz and trained subjects to report whether they are perceived as higher or lower in frequency than a standard at 440 Hz (two-interval task). The amplitude of each stimulus was different (i.e. values obtained from the green equal-intensity curve) in order to match perceived intensity. The chosen range of frequencies is the one where the required test and probe amplitudes can

be physically produced by the hardware available in our laboratory (P-841K191 actuator, E-618.1 high voltage amplifier and E-509.S1 controller, Physik Instrumente).

The subjects obtained visual feedback about correct performance continuously (red or green answer highlight after each trial). This training lasted between 80 and 250 trials (depending on subject's performance). Because all stimuli were intensity matched, intensity could not be used to perform the discrimination (only perceived pitch).

After training under these conditions, we introduced two probe standard amplitudes with amplitude change factors 1.67 and 0.77 of the test relative to the standard stimulus. These probe amplitudes occurred on 1/3 of the trials. No feedback about correct performance was given. The subjects completed between 500 and 600 in total during the session.

The obtained results (Supplementary Fig. 1A,B) for 6 tested subjects show consistent shifts in the psychometric curve and therefore exclude the possibility that they are due to differences in perceived intensity. For comparison purposes, the data is plotted on the same scale as the fits in the new Fig. 2B. The interpretation or comparison of these shifts is problematic because the test vibrations were each at different amplitude.

We repeated the same experiment for the 12 μm equal intensity curve, and once again observed the expected effect of amplitude on discrimination performance (Supplementary Fig. 1C,D).

Supplementary Figure 1. Pitch discrimination along equal intensity curves. **A:** Eight test stimuli (blue squares) were compared in perceived pitch to a standard stimulus (black square). Their frequencies were uniformly distributed around the 440 Hz standard and their amplitudes were obtained from the 10 μm equal intensity curve of Fig. 4A (i.e. all stimuli were matched in perceived intensity). Equal amplitude stimuli (circles) used in the experiment of Fig. 2 are shown for comparison. **B:** Psychometric curve fits (N=6 participants, individual panels) to the fraction of “higher” responses for vibrations with “matched” reference and test amplitudes (black), and for 2 tested amplitude change factors (yellow and blue). **C:** Same data as in B, for the same participants, for intensity matched stimuli according to the 12 μm equal intensity curve of Fig. 4A.

The following comments are taken from the follow-up discussion. Unfortunately we mistakenly wrote in our first email conversation that we will take stimuli along the “equal pitch” rather than the “equal intensity” curve, which led to a misunderstanding about the proposed control experiments. We hope the experiments and outcomes are clear now.

My main point is that the way the frequency discrimination tests in the paper were designed, it 'pays' for the subjects (both humans and mice) to use intensity as a cue. Because the green lines in Fig. 3B are not flat, intensity informs about frequency. If you have two stimuli that are close in frequency, picking the stronger one is on average likely to give you the one that is closer to your most sensitive frequency. This interpretation also explains why the size of the reported effect appears to depend on the local slope of the green lines, as this slope is a measure for the size of this additional, unintended cue. The control experiment now proposed by the authors prevents this effect to happen already during the training stage, but does not prevent it to develop during the testing stage.

Answer:

It is not clear how the effect of intensity can develop during the testing phase. All test stimuli are intensity matched and no feedback about correct performance is given (although this latter point should not matter). We simultaneously tested two probe stimuli, one with a decreased and one with an increased amplitude relative to the reference. We do not understand how it would be possible for the subjects to start basing their discrimination on perceived intensity during the course of the experiment.

The authors should therefore change their design and balance the stimuli for having the same range (and number) of perceived intensities at all frequencies. This new set of stimuli should all fall on a few green lines. This design would largely preclude using intensity as a useful cue during the test phase. According to the authors, there is a "fundamental relationship between the seemingly unrelated concepts of spectral sensitivity and pitch perception" and the magnitude of the shift should therefore stay the same. My prediction is that the effect will become smaller or may even disappear altogether in the test I propose here. I am curious to learn the outcome.

Answer:

As stated above we have carried out exactly these experiments. We suppose that an initial misunderstanding of the proposed control experiment occurred because (during the first email exchange) we mistakenly wrote that we will take stimuli along the “equal pitch” rather than the “equal intensity” curve. We are sorry for this confusion.

The human experiments described in Figure 1 were also not designed to stringently separate the intensity and frequency cues. During the instructions the difference between the two was verbally explained, but not systematically trained. Only easy trials differing strongly in frequency (and also in amplitude) were being trained. It is therefore no wonder that the intensity cues are not entirely disregarded, whereas Fig. 3 shows clearly that this cue can be perceived well, independently from the frequency cue.

Answer:

During training, all Aref amplitudes were tested simultaneously at the largest frequency differences between f_{test} and f_{ref} . Accordingly, A_{test} was always 11.8 μm but Aref took on all the values between 7.4 μm and 18.9 μm . The subjects were therefore not only instructed to ignore the amplitude but explicitly trained to do so.

Minor points:

I.32: "the main property of airborne vibrations (i.e. sound) is pitch perception" -> "the main distinguishing property of airborne vibrations (i.e. sound) is its pitch"

We thank the reviewer for this correction. However, based on the suggestion of Reviewer 3, this sentence was removed from the abstract.

I.42: "to" missing.

Corrected.

I.99 ostensibly

Corrected.

I.172: the shape of the reference curve in 1B depends on the test frequency, which suggests a Bayesian strategy of the mice: as the test stimuli get more difficult, the false-positive rate of the ref stimuli seems to go up as well.

Answer:

The reviewer seems to be referring to the fact that the black curves are not symmetric with respect to the middle of the tested frequency range. Due to our asymmetric task design (correct Go responses are rewarded and correct No-go responses are not) mice had a licking bias (i.e. a Go response bias). As a result, responses to Go frequencies were almost always at 100% correct and the point of subjective equality (i.e. 50% of lick responses) was not in the middle of the tested frequency range (vertical dotted line) and varied from day to day depending on the amount of licking bias. This bias did not compromise testing our prediction since the probe trials were intermingled with those of the trained amplitude in the same session.

I.252: The methods for the mouse tasks are somewhat unusual with a categorization task that is also a detection task (Fig. 1B, ACF=0.2), and which is implemented as a Go/NoGo task, whereas the detection threshold task is implemented as a two-alternative task.

Answer:

We agree with the reviewer that these methods are not standard. We initially used a Go/NoGo task for the frequency discrimination since it is more easily learned by mice than a 2AFC task. For consistency, we kept using this task thereafter (for the additional frequency locations). Given the observed

asymmetry and licking bias (see previous answer), we then chose to resort to a 2AFC task for the detection experiment.

I.258: the authors should check the equation for calculating P_{go} . It seems to involve negative numbers raised to a non-integer power.

We thank the reviewer for this correction. In fact, the indicated value for the S2 slope entails raising negative numbers to a non-integer power. This value was rounded in our code to the closest even integer. After checking the equation we also noticed that the correct definition for the bias value is the difference in the fraction of correct responses between no-go and go, and not the other way around.

I.310: Why a staircase procedure to obtain a psychometric curve?

The staircase procedure was simply used to increase the number of repetitions for stimuli for difficult comparisons and decrease the unnecessary repetitions for easy comparisons.

I.321: Why is this called a forced choice task if trials without a response were neither rewarded nor punished?

As stated in the methods, trials without a response constituted less than 5% of all trials and tended to occur towards the end of the session when the mice became satiated. 2AFC typically refers to a task where two alternative responses need to be given to two versions of a stimulus. This is the case in our experiment in more than 95% of trials. In such designs, especially involving animals, it is impossible to avoid aborted (without a response) trials. From our experience, punishing aborted trials does not decrease their occurrence. We agree with the reviewer that, in general, the 2AFC terminology is problematic because the choices are never really forced; the animal or human participant always has the option of not answering.

Discussion: Stevens' rule for the pitch shift is a small effect (about a semitone max, see the Cohen ref), and it has not always been easily reproduced. I'm not sure the superficial resemblances with the experiments in the present paper justify calling these a fundamental relationship between spectral sensitivity and pitch perception (I.22).

We thank the reviewer for this insight. We now nuance the description of Stevens' findings in the discussion by stating that "... although the size of the effect seems to be much smaller than initially reported (Cohen, 1961)." The "fundamental relationship between spectral sensitivity and pitch perception" statement refers, in the manuscript, only to our own results and not to the Stevens' study.

REVIEWERS' COMMENTS

Reviewer #2 (Remarks to the Author):

I have read through the revised manuscript and I thank the authors for their additional parameterization of the data.
I recommend publication.

Reviewer #3 (Remarks to the Author):

The authors have edited their manuscript to address my comments and suggestions, and those of the other 2 reviewers. I have no further suggestions to add, and I congratulate the authors on an interesting manuscript.

Reviewer #4 (Remarks to the Author):

In their revised manuscript Prsa et al have added an experiment to deal with the possibility that intensity cues were used to inform about frequency cues (Fig. S1). As the shift was still consistently present, the new experiment helps to rule out the possibility that intensity was employed as a cue for frequency in the test.
My other points have also been satisfactorily resolved.